# Molecular Characterization of Seasonal Influenza A and B from Hospitalized Patients in Thailand in 2018–2019

**DOI:** 10.3390/v13060977

**Published:** 2021-05-25

**Authors:** Kobporn Boonnak, Chayasin Mansanguan, Dennis Schuerch, Usa Boonyuen, Hatairat Lerdsamran, Kultida Jiamsomboon, Fanny Sae Wang, Arun Huntrup, Jarunee Prasertsopon, Nathamon Kosoltanapiwat, Pilaipan Puthavathana

**Affiliations:** 1Department of Microbiology and Immunology, Faculty of Tropical Medicine, Mahidol University, Bangkok 10400, Thailand; kobporn.boo@mahidol.ac.th (K.B.); dennis.schuerch@gmail.com (D.S.); kultidacare@gmail.com (K.J.); fanny.w.841119@gmail.com (F.S.W.); nathamon.kos@mahidol.ac.th (N.K.); 2Department of Clinical Tropical Medicine, Faculty of Tropical Medicine, Mahidol University, Bangkok 10400, Thailand; chayasin.man@mahidol.ac.th; 3Department of Molecular Tropical Medicine and Genetics, Faculty of Tropical Medicine, Mahidol University, Bangkok 10400, Thailand; usa.boo@mahidol.edu; 4Center for Research and Innovation, Faculty of Medical Technology, Mahidol University, Nakhon Pathom 73170, Thailand; hatairat.ler@mahidol.ac.th (H.L.); jarunee.pra@mahidol.ac.th (J.P.); 5Hospital for Tropical Diseases, Faculty of Tropical Medicine, Mahidol University, Bangkok 10400, Thailand; arun.hut@mahidol.ac.th

**Keywords:** seasonal influenza, epidemiology

## Abstract

Influenza viruses continue to be a major public health threat due to the possible emergence of more virulent influenza virus strains resulting from dynamic changes in virus adaptability, consequent of functional mutations and antigenic drift in surface proteins, especially hemagglutinin (HA) and neuraminidase (NA). In this study, we describe the genetic and evolutionary characteristics of H1N1, H3N2, and influenza B strains detected in severe cases of seasonal influenza in Thailand from 2018 to 2019. We genetically characterized seven A/H1N1 isolates, seven A/H3N2 isolates, and six influenza B isolates. Five of the seven A/H1N1 viruses were found to belong to clade 6B.1 and were antigenically similar to A/Switzerland/3330/2017 (H1N1), whereas two isolates belonged to clade 6B.1A1 and clustered with A/Brisbane/02/2018 (H1N1). Interestingly, we observed additional mutations at antigenic sites (S91R, S181T, T202I) as well as a unique mutation at a receptor binding site (S200P). Three-dimensional (3D) protein structure analysis of hemagglutinin protein reveals that this unique mutation may lead to the altered binding of the HA protein to a sialic acid receptor. A/H3N2 isolates were found to belong to clade 3C.2a2 and 3C.2a1b, clustering with A/Switzerland/8060/2017 (H3N2) and A/South Australia/34/2019 (H3N2), respectively. Amino acid sequence analysis revealed 10 mutations at antigenic sites including T144A/I, T151K, Q213R, S214P, T176K, D69N, Q277R, N137K, N187K, and E78K/G. All influenza B isolates in this study belong to the Victoria lineage. Five out of six isolates belong to clade 1A3-DEL, which relate closely to B/Washington/02/2009, with one isolate lacking the three amino acid deletion on the HA segment at position K162, N163, and D164. In comparison to the B/Colorado/06/2017, which is the representative of influenza B Victoria lineage vaccine strain, these substitutions include G129D, G133R, K136E, and V180R for HA protein. Importantly, the susceptibility to oseltamivir of influenza B isolates, but not A/H1N1 and A/H3N2 isolates, were reduced as assessed by the phenotypic assay. This study demonstrates the importance of monitoring genetic variation in influenza viruses regarding how acquired mutations could be associated with an improved adaptability for efficient transmission.

## 1. Introduction

Seasonal influenza virus epidemics are a source of respiratory disease burden which results in substantial numbers of deaths each year [1]. Four co-circulating subtypes/lineages of influenza viruses currently cause disease in humans; A/H3N2, A/H1N1, B/Victoria/2/87-like, and B/Yamagata/16/88-like viruses [2]. The rapid evolution and accumulation of amino acid changes in the surface protein, hemagglutinin (HA), and neuraminidase (NA), results in escape from neutralizing antibodies and is the major cause of antigenic drift [3,4]. Influenza viruses have evolved multiple strategies to interact with the host in order to complete their life cycle. Several intensive studies have revealed that the evolution of influenza viruses are mainly mediated through the mutation of the virus itself and the re-assortment of viral genomes derived from various influenza strains. Importantly, the evolution of influenza viruses through these mechanisms results in worldwide annual epidemics and occasional pandemics [5,6,7,8]. The emergence of antigenic drift influenza viruses necessitates updates in vaccine composition to ensure optimal antigenic characteristics of annual seasonal influenza vaccines. The World Health Organization (WHO) annually conducts global surveillances of influenza, and predicts the representative influenza virus strains for vaccine development [9]. However, the lag time between virus identification and vaccine distribution exceeds 6 months; thus, the vaccine is sometimes unable to provide immediate protection against sudden influenza outbreaks [10]. The monitoring of severe influenza cases is essential for evaluating influenza virus evolution and detecting the emergence of new variants of influenza virus that may pose a pandemic threat. Considering the high mutation rate of influenza viruses, genomic surveillance and determining the molecular patterns of influenza and its circulation needs to be conducted more systematically and regularly. As part of this effort, we conducted a molecular analysis of viruses isolated from severe influenza cases in Thailand. The whole genome sequences of the influenza viruses isolated in this study were characterized in comparison with the vaccine strains and other influenza genome sequences from public databases. Moreover, the susceptibility of these influenza viruses to oseltamivir neuraminidase-inhibitor was also determined by neuraminidase-inhibition (NAI) assay. Our finding may help to identify viral factors for disease severity in a broader sense, and further provide insights on Thailand’s influenza isolates in the context of global influenza circulation.

## 2. Materials and Methods 

### 2.1. Ethical Considerations

This study was approved by the Mahidol University Central Institutional Review Broad (MU-CIRB) Thailand on 10 January 2018 (Approval number: MU-CIRB 2018/014.1601).

### 2.2. Specimen Collection and Influenza Virus Detection

Sputum samples were collected from hospitalized patients (between 18 and 65 years old) in the Bangkok Hospital for Tropical Diseases, Faculty of Tropical Medicine, Mahidol University, who presented signs and symptoms of a severe respiratory infection and tested positive for influenza via the QuickNaviTM Flu + RSV rapid test kit (Denka Seiken, Japan). The sputum samples were kept in universal transport medium (UTM) (COPAN, Brescia, Italy) per the manufacturer’s manual for further laboratory tests. 

### 2.3. Virus Isolation 

Viruses from each specimen were isolated in Madin-Darby Canine Kidney (MDCK) cells (American Type Culture Collection; CCL-34). In brief, sputum samples in UTM were vigorously vortex and centrifuged at 3000 rpm for 10 min and the supernatant was collected and kept on ice. Approximately 100 µL of supernatant was added into 80 to 90% confluent monolayer of MDCK cells in 400 µL Minimum Essential Medium Eagle (MEM) (Gibco, life Technologies, Grand Island, NY, USA) media supplemented with L-glutamine, antibiotic-antimycotic (Invitrogen-Gibco, Carlsbad, CA, USA), and 1 µg/mL trypsin-tosyl phenylalanyl chloromethyl ketone (Trypsin-TCPK) (Sigma Aldrich, St. Louise, MO, USA). The infected MDCK monolayer was observed daily for the appearance of cytopathic effect (CPE). Cell culture containing viruses were harvested, centrifuged and hemagglutination assay was performed to determine the hemagglutinin titer of the cultured supernatants. The supernatants containing the virus was aliquoted and stored at −80 °C for further experiments. The positive cultured viruses were passaged up to three times to obtain sufficient virus titers to perform virus identification, NAI assay, and whole genome sequencing. 

### 2.4. Hemagglutination (HA) Assay

To assess the titer of isolated viruses, hemagglutination assay was performed as previously described [11]. The test virus was twofold serially diluted with phosphate buffer saline (PBS) in a volume of 50 µL/well in duplicate. Fifty microliters of 0.5% goose or 0.75% guinea pig erythrocyte suspended in PBS was added into the test wells and incubated for 30–45 min at room temperature before determining the hemagglutinating results. The hemagglutination unit of the test virus was defined as the highest virus dilution to display complete hemagglutination activity.

### 2.5. Viral RNA Extraction, Influenza Subtype Determination, and Genome Sequencing

Viral RNA was extracted from sputum samples or viral cultured supernatants using AccuPrep^®^ Viral RNA Extraction Kit (BioNEER Corp, Korea) according to the manufacturer’s instructions. One-step RT-PCR was performed to differentiate A/H1N1, A/H3N2, and influenza B based on WHO recommended protocol [12]. Sanger sequencing was employed to obtained the nucleotide sequences of all 8 viral segments. Nucleotide sequences from different genome segments were assembled into contigs using the Bioedit software (version 7.2.5; https://bioedit.software.informer.com, accessed on 10 January 2020). The nucleotide sequences described in this study have been deposited into the GenBank database (Appendix A).

### 2.6. Phylogenetic Analysis

For each influenza gene segment of A/H1N1, A/H3N2, and influenza B viruses, a phylogenetic tree was constructed with the nucleotide sequences identified in this study, WHO recommended vaccine strains, and full genome influenza virus sequences during 2018–2019, retrieved from the GenBank and GISAID database. The phylogenetic trees were constructed based on the Maximum-likelihood method using Tamura-3-parameter (T92) in the MEGA 6 software [13].

### 2.7. Sensitivity to Anti-Neuraminidase Drugs

A fluorescence-based NA inhibition (NAI) assay was used to determine the sensitivity of influenza viruses to the viral NA inhibitor as previously described [14]. Briefly, the assay principle is based on the measurement of 4-methylumbelliferone (MU), the fluorescent product obtained after cleavage of 2′2′-(4-Methylumbelliferyl)-α-D-N-acetylneuraminic acid (MUNANA) substrate by the viral NA. In the presence of the NA inhibitor, viral NA activity will be blocked and result in the reduction of MU product. In this study, oseltamivir carboxylic acid (Roche, Basel, Switzerland) was serially fourfold diluted to obtain a final concentration of 4000 to 0.015 nM in a 10 µL. Thereafter, the test virus at the standard dose in a 10 µL volume was added into each well, mixed and incubated at 37 °C with shaking in the dark for 30 min. Subsequently, 30 µL of MUNANA substrate was added and the reaction plate was further incubated at 37 °C with shaking for 60 min in the dark. The reaction was terminated by adding stop solution. The experiments were performed in duplicate. Fluorescence intensity of MU was measured, and 50% inhibitory concentration (IC50) value was determined from the dose-response curve. The IC50 is defined as the amount of drug that decreases the amount of MU produced for 50%. The NA inhibitor- sensitive virus will show a low IC50 value in the NA inhibition assay. On the other hand, viruses with genetic changes in the NA gene may produce a neuraminidase protein that is resistant to the drug, leading to a high IC50 value. 

### 2.8. Homology Modelling

Three-dimensional (3D) structural model of the H1N1 protein structure was constructed and its putative interactions with sialic acid were predicted. The impact of mutations on sialic acid binding site was illustrated. The structure of influenza hemagglutinin from the A/California/07/2009 (H1N1) (PDB ID: 3UBE), A/Victoria/361/2011 (H3N2) (PDB ID: 4O51), and B/Brisbane/60/2008 (PDB ID:4FQM) were used as a template for protein modeling. The model was constructed using SWISS-MODEL (https://swissmodel.expasy.org, accessed on 15 January 2020) [15] and the structural model was verified by PROCHECK [16]. Then, the derived structures were complexed with sialic acid and evaluated by AutoDock Vina [17]. The structural models were visualized by Discovery Studio Visualizer-Accelrys (https://discover.3ds.com, accessed on 15 January 2020).

## 3. Results

### 3.1. Patient Demographic Data

The median age of influenza infected patients was 52 years old. Female accounted for 76.7% of influenza infected patients. In all diagnosed severe cases, approximately 80% of these cases had a chronic medical condition. Regarding the age distribution data for all diagnosed cases, 43% of severe cases were associated with those aged 60 or older (Table 1). In the current study, severe influenza cases were associated with A/H1N1, A/H3N2, and influenza B for patients of all ages.

### 3.2. Genome Characterization of Influenza Virus Isolates

Of 30 laboratory confirmed cases, the influenza type/subtype identified comprised 9, (30%), 14 (46.7%), and 7 (23.3%) of A/H1N1, A/H3N2, and influenza B, respectively. Out of 30 sputum samples, 20 influenza virus isolates were obtained and subjected to molecular-genetic analysis. We identified 7 A/H1N1, 7 A/H3N2, and 6 B/Victoria viruses. The identification numbers, strain names, collection dates, and GenBank accession numbers of all 20 viruses detected in this study are shown in Appendix A and the hemagglutination titers of these isolated viruses are shown in Appendix A.

### 3.3. Evolutionary Characteristic and Structure Variations in the HA Domain of H1N1 Viruses

To better understand the evolutionary trend of A/H1N1 viruses in Bangkok, Thailand, during the period of 2018–2019, all gene sequences of the seven influenza A (H1N1) isolates from our study were analyzed alongside the global strains. The phylogenetic tree concatenates from all eight segments of the influenza A/H1N1 strains (Figure 1), as well as the the phylogenetic trees for the other segments of each virus subtype (Appendix A) were constructed. Five out of seven A/H1N1 isolates clustered together with the viruses belonging to subclade 6B.1, particularly closest to the A/Switzerland/3330/2017 (H1N1), whereas the other two isolates clustered with A/Brisbane/02/2018 (H1N1) (recommended composition of influenza vaccine for use in 2020 Southern Hemisphere influenza season) which belongs to subclade 6B.1A1 (Figure 1). These results indicated that some of the circulating A/H1N1 viruses were beginning to drift from the A/H1N1 vaccine strain given in 2018–2019.

The homology-based modeled HA mutated structure in comparison with A/California/07/2009 (H1N1) revealed that the observed mutation among H1N1 strains in our study are unlikely to alter the overall tertiary structure of HA (Figure 2A). Substitution at positions P100S, S101N, D114S, S179N, K180Q, S181T, T214A, S220T, and I233R were detected in the head domain of H1N1 strain isolates in this study. These mutations are common among the A/H1N1 global strains reported after pandemic 2009. Among these Thai strains, we found additional mutations at antigenic sites; S91R (Cb), S181T(Sa), and T202I (Sb) (Figure 2B). Moreover, a unique mutation (S200P) was observed at the receptor binding site (Figure 3A). In fact, serine at amino acid position 200 and asparagine 204 together coordinate a water molecule that hydrogen bonds with the 9-hydroxyl group of sialic acid. The mutation S200P results in the loss of coordination with a water molecule, and may interfere with the hydrogen bond to the Sialic molecule (Figure 3B). Determination of amino acid substitutions among the A/H1N1 isolates also revealed the mutation Q51K, F74S, G77R, V81A, I188T, and T462I for segment NA, T80A, A155T, and A242T for segment NS, V431I for segment NP, V200I and K386R for segment PB1, and G225S and V667I for segment PB2 (Appendix A). 

### 3.4. Evolutionary Characteristic and Structure Variations in the HA Domain of H3N2 Viruses

The phylogenetic tree from the concatenated segment from the seven influenza A/H3N2 strains (Figure 4), as well as the the phylogenetic trees for each segment of the virus subtype were constructed (Appendix A). The isolates TM-54_54, TM-5434_51, and TM-8453_54 clustered into subclade 3C.2a2, represented by the WHO vaccine strain A/Switzerland/8060/2017 (H3N2) for 2018. The isolates TM-358_60, TM-12909_43, TM-12054_56, and TM-17617_61 fell into subclade 3C.2a1b, represented by the newly announced WHO vaccine strain, A/South Australia/34/2019 for 2020 Southern Hemisphere season. The homology based modeled HA mutated structure in comparison with A/Switzerland/8060/2017 (H3N2) revealed that the observed mutations among H3N2 strains in our study are unlikely to alter the overall tertiary structure of HA (Figure 5A,B). These isolates contain substitutions of T144A/I, T151K at antigenic site A, Q213R, S214P, and T176K at antigenic site B, D69N and Q277R at antigenic site C, N137K and N187K at antigenic site D and E78K/G at antigenic site E (Figure 5C). The isolates of the 3C.2a1b clade also contain amino acid substitutions at P126L, K220N, and V303I in NA, K158R in PA, and S107N in PB2. Isolates of the clade 3C.2a2 possess methionine and serine at NA residues 176 and 386, respectively (Appendix A).

### 3.5. Evolutionary Characteristic and Structure Variations in the HA Domain of Influenza B Viruses

All six influenza B isolates clustered in clade 1A3-DEL of the Victoria lineage and are closely related to B/Washington/02/2009 (Figure 6 and Appendix A). This clade is characterized by a three amino acid deletion on the HA protein at positions K162, N163, and D164. As isolate TM-3038_62 does not possess the 3 amino acid deletions, it falls into subclade 1A. Among the isolates clustered within the 1A3-Del subclade, we observed similar amino acid residues as the newly introduced representative strain, B/Washington/02/2019. In comparison to the B/Colorado/06/2017 vaccine strain, these substitutions include G129D, G133R, K136E, and V180R for segment HA (Figure 7). Additional amino acid substitutions were found at P99S, C123S, and S139C for segment NA, T285I for segment M, and V453A at the NP segment (Appendix A).

### 3.6. Analysis of Sensitivity to Anti-Neuraminidase Drugs

Sensitivity to oseltamivir of the current influenza virus isolates were assessed by NAI assay. The results showed that all influenza A/H1N1 and A/H3N2 isolates were sensitive to the anti-neuraminidase drug, oseltamivir. All strains of these viruses showed sensitivity to oseltamivir within the normal IC50 range of 0.1–1.5 nM [18,19]. However, the influenza B isolates were found to have reduced sensitivity to oseltamivir with IC50 values ranging from 13.34 to 42.68 (Table 2).

## 4. Discussion

To better understand the evolutionary trend of the current circulating influenza viruses in Thailand, the whole genome sequences of influenza A/H1N1, A/H3N2, and influenza B isolates from our study (Appendix A) were analyzed along with the global strains. The A/H1N1 Thai strains were found to have similar mutations as those reported in global A/H1N1 isolates worldwide [20]. In general, all Thai A/H1N1 strains fell into the sub-clade 6B.1 and were genetically similar to the recommended A/H1N1 vaccine strains [21]. However, two of the Thai A/H1N1 isolates clustered with the newly announced 2020 Southern Hemisphere vaccine component, A/Brisbane/02/2018, representing sub-clade 6B.1A1. Alignment of HA amino acid sequences of the Thai A/H1N1 virus isolates in the current study showed several amino acid changes when compared to A/California/07/2009 (H1N1). We observed similar mutations with the A/H1N1 isolates reported from India in 2017 with additional mutations at antigenic sites Sa, Sb, and Cb [22,23,24]. It is unknown whether these substitutions confer to the virus’s fitness, as such, further research for determining the role of these changes is recommended. We also found a mutation at the receptor binding site (S220T) as previously reported in the Indian isolates [25] which suspected that this mutation would increase the receptor-binding avidity [26]. However, an additional mutation (S200P) was detected in the Thai A/H1N1 isolates. Molecular protein docking revealed that this substitution position may alter the receptor binding avidity but further investigation is needed for confirmation. 

Whole genome sequencing revealed that the H3N2 isolates belonging to genetic clades 3C.2a2 and 3C.2a1b, could be represented by the Southern Hemisphere vaccine strains A/Switzerland/8060/2017(H3N2) and the newly announced A/South Australia/34/2019 (H3N2) [21], respectively. Currently, the H3N2 vaccine component for the 2019 Southern Hemisphere includes A/Switzerland/8060/2017 (clade 3C.2a2) which has not been included in 2018. In light of this, the 2018–2019 Thailand isolates that have clustered within clade 3C.2a2 and 3C.2a1b suggest a lack of vaccine coverage in Thai people. Similarly, with regards to a newly announced vaccine component, WHO has recommended the B/Washington/02/2019 vaccine strain (clade 1A 3-Del) to be included as the Victoria lineage component for the 2020 Southern Hemisphere vaccine. In this study, five of the six Flu B isolates revealed a mismatch to the current vaccine component, B/Colorado/06/2017, which represents clade 1A 2-Del [27]. Rather, it was observed that the study isolates presented similarities to the B/Washington/02/2019 strain. This indicates that an update in the Victoria vaccine component was necessary to account for the rise in circulating 1A 3-Del strains. However, one out of six Flu B isolates in this study did not cluster within the 1A 3-Del clade as the other isolates, which may indicate a co-circulating of influenza B variants in Thailand. Further, it should be noted that no Yamagata isolates were included in the study. 

None of the A/H1N1 isolated viruses were sensitive to neuraminidase inhibitors. The sequence analysis of the NA gene reveals one potential NA drug-resistance substitution (V116A) but not other amino acid substitutions such as I117V, Q136K, D151A, Y155H, R156K, D198V, I222R, R224K, Q226H, E227D, E227Q, H274Y, R293K, N294S, E425G, and I436N [28,29]. The other six amino acid differences (Q51K, F74S, G77R, V81T, and T462I) for segment NA relative to A/Brisbane/02/2018 (H1N1) were found in our A/H1N1 isolates. These two mutations (Q51K and F74S) were similar to the recent isolated A/H1N1 viruses from Korea, but the substitution functionality remains unknown [29]. For H3N2 viruses, we did not detect the mutations conferring resistance to neuraminidase inhibitors such as E119V, D151E, I222V, R224K, E276D, N249S, R292K, and R371K [30] in the NA gene segment of our H3N2 isolates. We found three additional amino acid differences (P126L, K220N, and V303I) in the NA gene of our A/H3N2 isolates relative to A/Kansas/14/2017 (H3N2). Among these additional substitutions, the V303I amino acid substitution has been reported in the A/H3N2 viruses with a low resistance to neuraminidase inhibitors [31]. In agreement with other studies [32,33,34,35], all of our Flu B isolates reduced susceptibility to neuraminidase inhibitors. Our Flu B isolates contain amino acid substitutions at R152K, D198N, and R371K, which have been reported to associate with the neuraminidase inhibitor resistance phenotype of influenza B [36]. We also found additional mutations such as P99S, C123S, and S139 at NA segments. However, the contribution of these mutations to resistance to neuraminidase inhibitor and their impact on viral fitness has not been reported, which warrants further investigating on viral replication in vitro and in vivo as well as transmission of the variants. The establishment of neuraminidase-inhibitor resistant, which is commonly found among influenza B viruses would be a potential public health concern, especially when the neuraminidase inhibitors such as oseltamivir and zanamivir are currently the only available therapeutic options for influenza infections. Many aspects of the molecular markers confer resistance to neuraminidase inhibitor. Frequency of emergence and fitness of neuraminidase-resistant variants of influenza B are far less studied than influenza A. 

A few limitations of the study may have affected the accurate representation of circulating serotypes. Firstly, a small sample size may not be adequate to make definitive conclusions. Secondly, the samples were collected only in Bangkok, which may not represent the epidemiology of influenza viruses for the entire country. Thirdly, most patients in our study had an underlying health condition that increases the risk of severity. Thus, further study with a larger sample size in a healthy population is needed with sputum collected from major regions around Thailand. This would then allow a more accurate representation of the circulating strains. With regards to the severity of the cases, it is not possible to conclude which amino acid changes confer the virus’s ability to infect and replicate. For instance, amino acid changes could confer stronger immunodominance when favorable structural changes in antigenic sites occur due to changes in the molecular bonds; changes could also confer no structural change but contribute to another pathogenesis mechanism. Whole genome sequencing allows a full analysis of the virus’s genome and serves as a basis for further research to be conducted. Ultimately the severity may be a result of host factors such as age, immunocompromised state, vaccine coverage, or genetic susceptibility [37].

Finally, with the current trend of pandemics occurring approximately every 39–40 years [38], whole genome analysis of seasonal influenza viruses is an important surveillance system to identify a novel virus. Novel strains may be the result of genetic reassortment, combining viral gene segments from zoonic hosts and humans to produce a strain which humans lack immunity to, much like the case of H3N2 swine variant [39]. As such, the continual surveillance of novel strains is vital to the pandemic preparedness. 

## Figures and Tables

**Figure 1 viruses-13-00977-f001:**
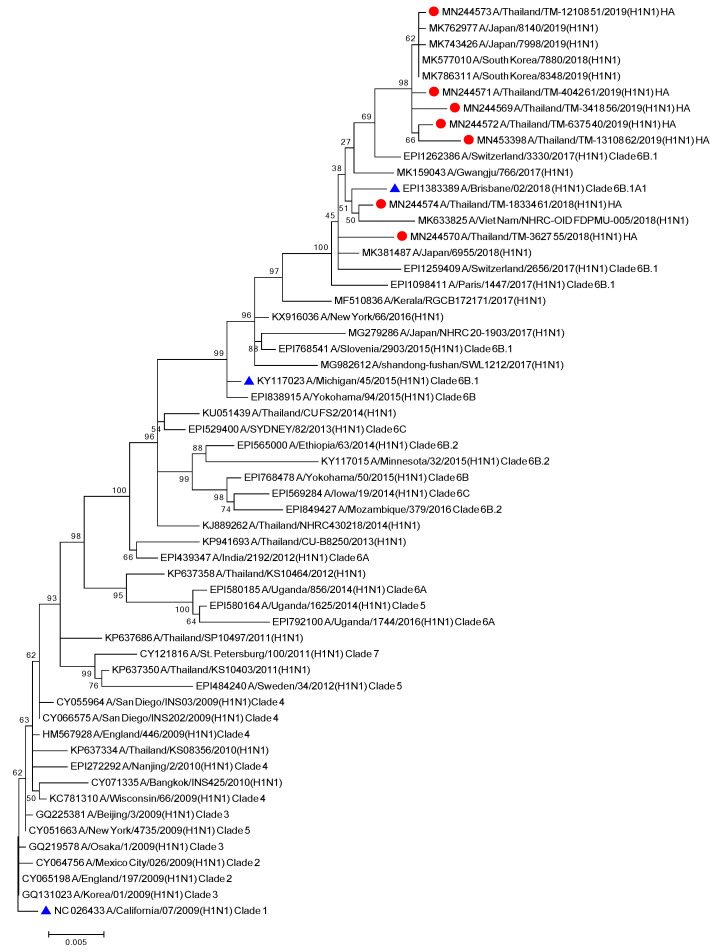
Phylogenetic tree of concatenate from all 8 genes of influenza A/H1N1 Thai strains was inferred by using the Maximum Likelihood method based on the Tamura 3-parameter model. Red circles indicate A/H1N1 Thai isolates and blue triangles indicate A/H1N1 vaccine strains.

**Figure 2 viruses-13-00977-f002:**
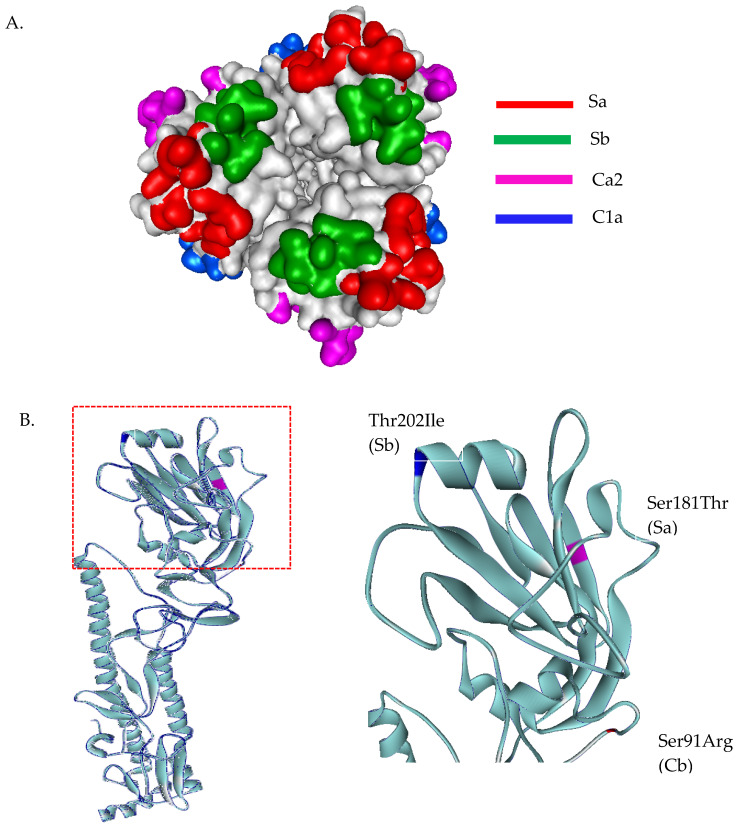
Three-dimensional structural analysis of A/H1N1 HA protein (**A**). Top view of head domain of H1N1 showing antigenic sites located in different regions of the HA molecule (**B**). Superimposed HA structures of A/Michigan/45/2015 (H1N1) and A/H1N1 Thai isolates, where mutations at antigenic sites are highlighted in blue, magenta, and red. HA structures were built based on the structure of A/California/07/2009(H1N1) HA protein, PDB ID: 3UBE.

**Figure 3 viruses-13-00977-f003:**
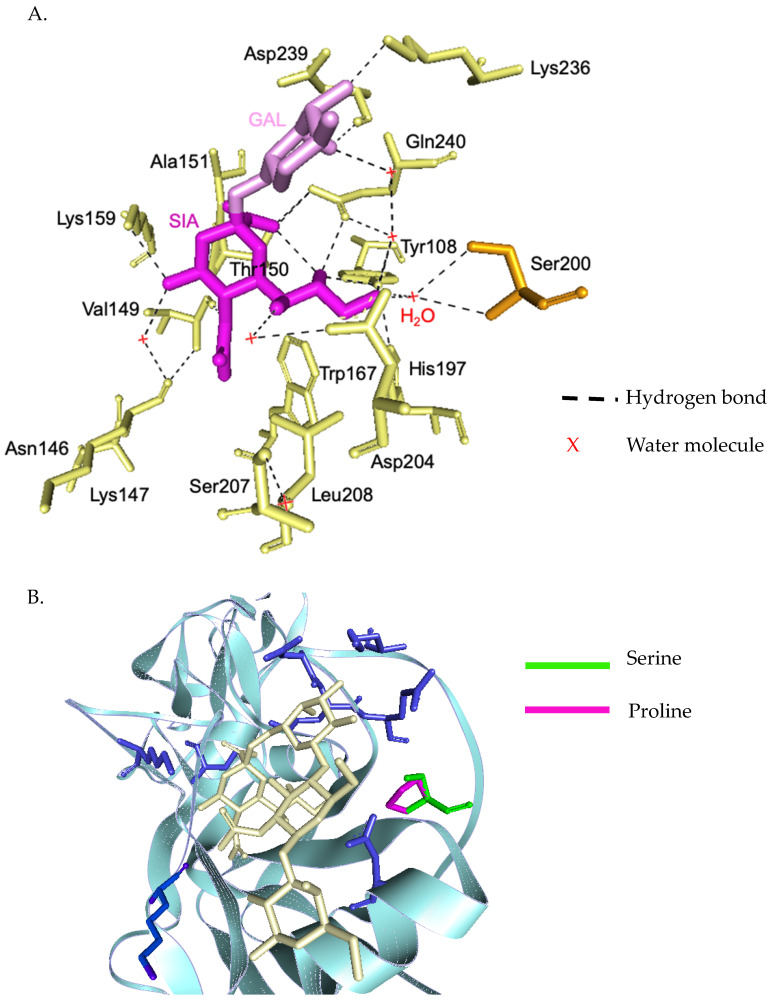
Three-dimensional structural analysis of H1N1 HA receptor binding site and interactions with the sialic acid receptor analogues (**A**). Residues and side-chain interactions of receptor binding site of A/Michigan/45/2015 (H1N1) HA protein and sialic acid. Ser200 and Asp204 together coordinate a water molecule that hydrogen bonds with the 9-hydroxyl group of sialic acid (**B**). Superimposed HA structures of A/Michigan/45/2015 (H1N1) and the mutant Ser200Pro. This results in a loss of coordination with water molecule, disrupting the hydrogen bond with sialic molecule.

**Figure 4 viruses-13-00977-f004:**
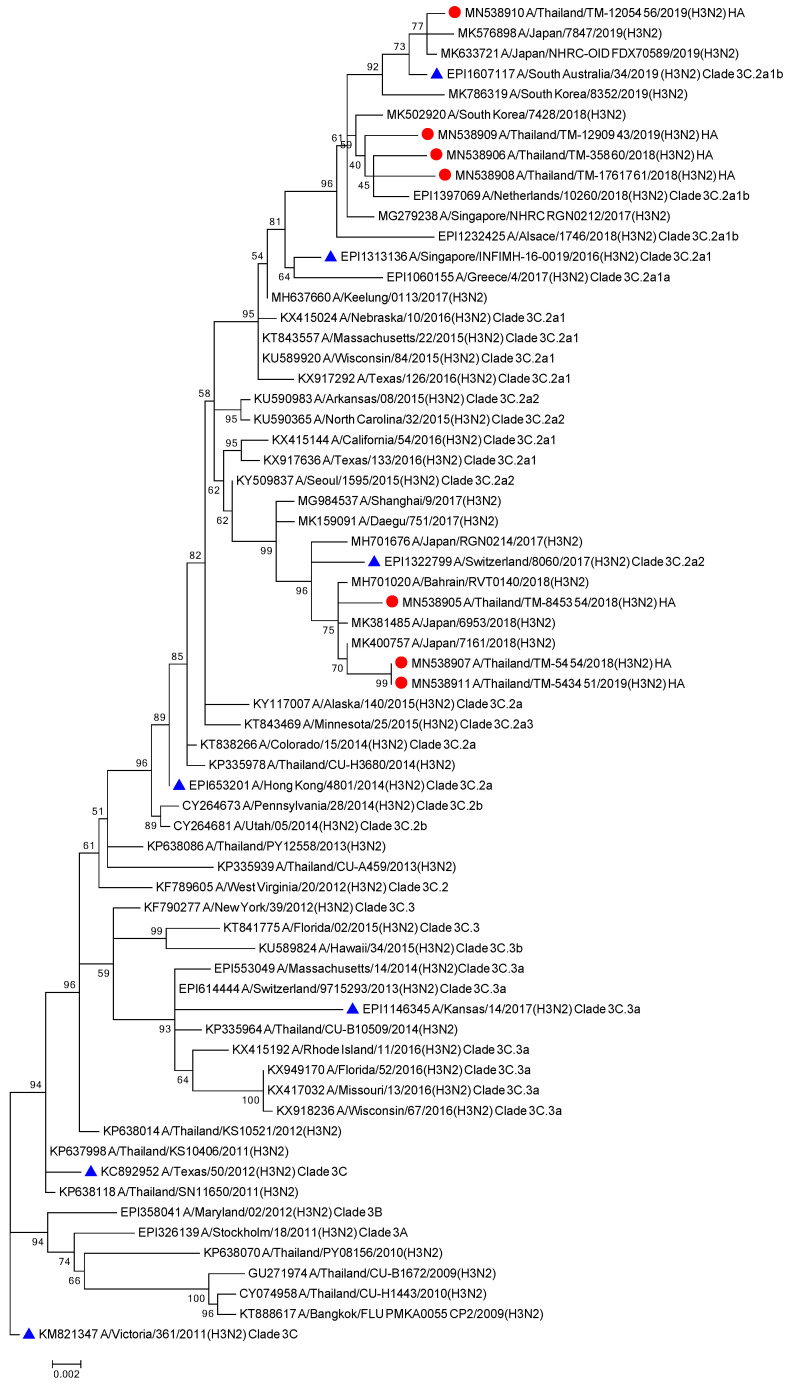
Phylogenetic tree of concatenate from all 8 genes of influenza A/H3N2 Thai strains was inferred by using the Maximum Likelihood method based on the Tamura 3-parameter model. Red circles indicate A/H3N2 Thai isolates and blue triangles indicate A/H3N2 vaccine strains.

**Figure 5 viruses-13-00977-f005:**
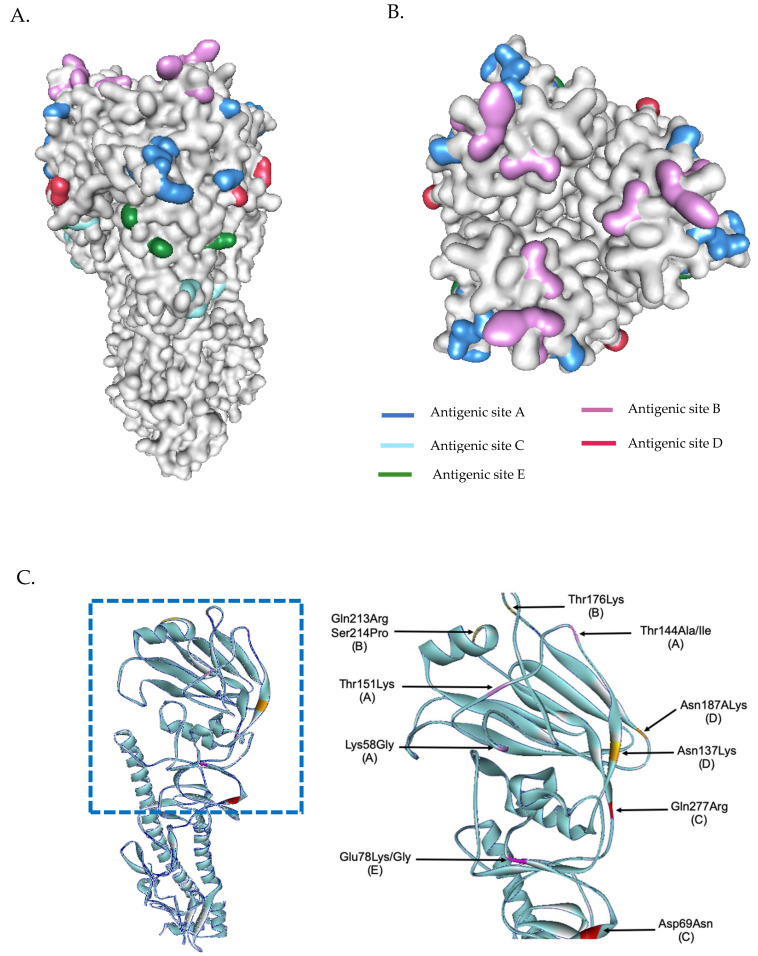
Three-dimensional structural analysis of H3N2 HA protein (**A**). Surface representation of H3N2 showing antigenic sites located in different regions of the HA molecule (**B**). Superimposed HA structures of A/Switzerland/8060/2017 (H3N2) and other isolates (or mutations) where mutations at antigenic sites are highlighted (**C**). HA structures were built based on the structure of A/Victoria/361/2011 (H3N2) influenza hemagglutinin protein (PDB ID: 4O5I).

**Figure 6 viruses-13-00977-f006:**
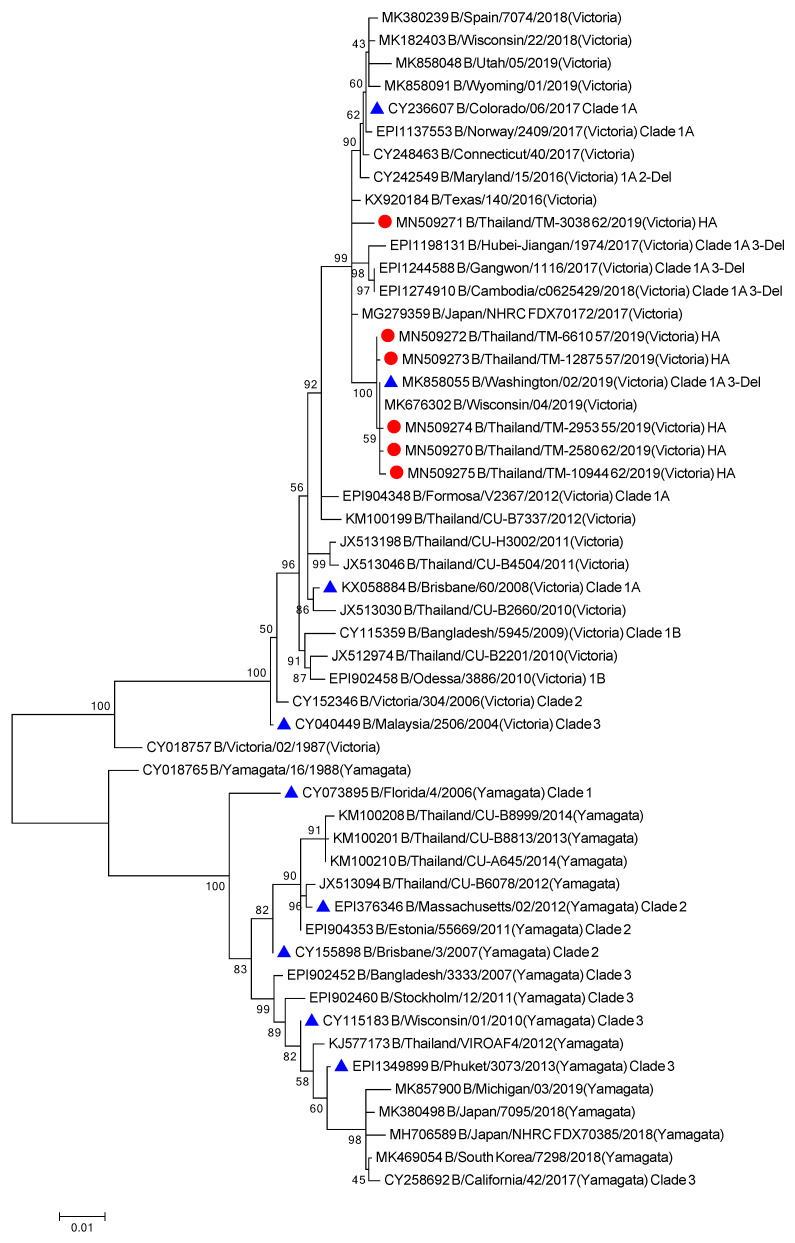
Phylogenetic tree of concatenate from all 8 genes of influenza B/Vic Thai strains was inferred by using the Maximum Likelihood method based on the Tamura 3-parameter model. Red circles indicate B/Victoria Thai isolates and blue triangles indicate influenza B vaccine strains.

**Figure 7 viruses-13-00977-f007:**
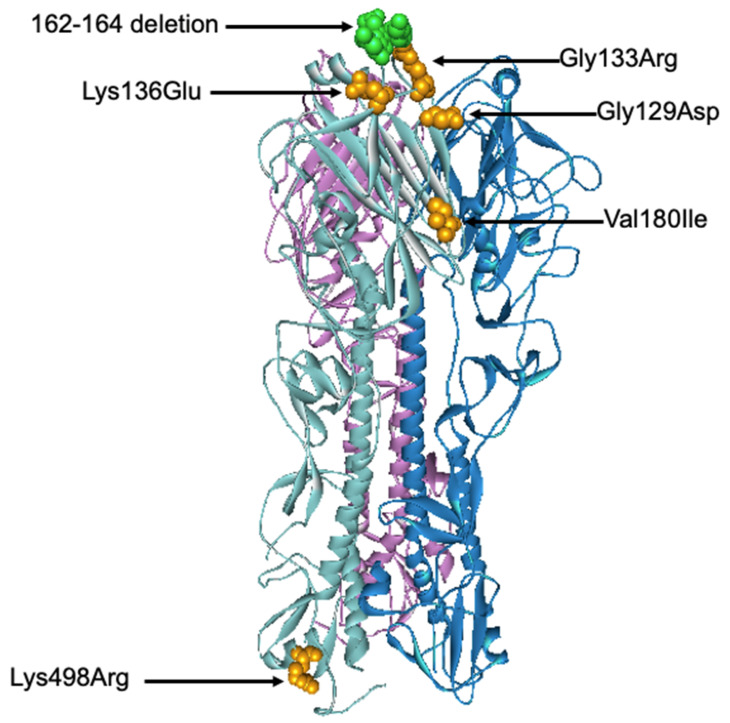
Three-dimensional structural analysis of the trimeric B/Colorado/06/2017 HA protein. Amino acid substitutions and deletion found in this study were highlighted in orange and green, respectively. The structure of B/Brisbane/60/2008 Influenza Hemagglutinin protein (PDB ID: 4FQM) was used as a template.

**Table 1 viruses-13-00977-t001:** Clinical characteristics of influenza associated hospitalization during 2018–2019 season (n = 30).

Characteristics	Number of Cases (n)
**Gender**	
Male	7
Female	23
**Age groups**	
15–35	7
36–45	4
46–59	6
≥60	13
**Co- morbidities**	
Cardiovascular diseases	2
Respiratory diseases	3
Chronic renal diseases	2
Diabetes mellitus	4
Hypertension	8
Cancer and hematological diseases	2
Stroke and neuromuscular diseases	3
**Days in ICU**	
1–5 days	0
>5 days	3
**Days in medical ventilation**	
1–5 days	0
>5 days	3
**Outcome**	
Death	0
Discharged	30

**Table 2 viruses-13-00977-t002:** Drug susceptibility testing by fluorescence-based neuraminidase inhibition assay.

Viruses	Subtypes	Passage No. of Isolate	Mean IC50 ± SD (nM)
A/Thailand/TM-18334_61/2018	H1N1	1,2,3	0.59 ± 0.07
A/Thailand/TM-3627_55/2018	H1N1	5,6,7	0.29 ± 0.08
A/Thailand/TM-12108_51/2019	H1N1	7,8,9	0.23 ± 0.05
A/Thailand/TM-6375_40/2019	H1N1	3,4,5	0.48 ± 0.07
A/Thailand/TM-4042_61/2019	H1N1	3,4,5	0.16 ± 0.01
A/Thailand/TM-3418_56/2019	H1N1	2,3,4	0.35 ± 0.04
A/Thailand/TM-13108_62/2018	H1N1	1,2,3	0.57 ± 0.01
B/Thailand/TM-6610_57/2019	B/Vic	3,4,5	33.96 ± 4.12
B/Thailand/TM-3038_62/2019	B/Vic	1,2,3	40.10 ± 0.91
B/Thailand/TM-12875_57/2019	B/Vic	2,3,4	23.38 ± 2.76
B/Thailand/TM-2580_62/2019	B/Vic	3,4,5	17.18 ± 3.38
B/Thailand/TM-2953_55/2019	B/Vic	1,2,3	36.84 ± 5.28
B/Thailand/TM-10944_62/2019	B/Vic	2,3,6	22.62 ± 1.15
A/Thailand/TM-8453_54/2018	H3N2	2,3,4	0.41 ± 0.04
A/Thailand/TM-17617_61/2018	H3N2	6,7,8	0.14 ± 0.02
A/Thailand/TM-12054_56/2019	H3N2	8,9,10	0.21 ± 0.04
A/Thailand/TM-12909_43/2019	H3N2	4,5,6	0.11 ± 0.02
A/Thailand/TM-358_60/2018	H3N2	7,8,9	0.05 ± 0.01
A/Thailand/TM-54_54/2018	H3N2	6,7,8	0.11 ± 0.01
A/Thailand/TM-5434_51/2019	H3N2	5,6,7	0.06 ± 0.01
A/Mississippi/03/01 *wt* (274H)	H1N1	6	0.67 ± 0.03
A/Mississippi/03/01mutant 274Y	H1N1	6	286.01 ± 14.02
A/Fukui/45/04 *wt* (119E)	H3N2	5	0.23 ± 0.02
A/Fukui/45/04 mutant (119V)	H3N2	5	30.66 ± 0.88

## Data Availability

Data is contained within article and Appendix A.

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
