# Peer review of "Molecular Characterization of Seasonal Influenza A and B from Hospitalized Patients in Thailand in 2018–2019"

_viruses, 2021, doi:10.3390/v13060977_

Round 1

Reviewer 1 Report

The manuscript presents an analysis of influenza viruses isolated from patients and compares the viruses to prior vaccine strains and contemporary viruses. The analysis includes an assessment of neuraminidase inhibitor susceptibility. In general the results indicate that the viruses that circulated in Thailand in 2018-2019 were similar to those that circulated elsewhere.  Although there are no exciting revelations resulting from this work, a better understanding of how influenza viruses evolve and where they circulate globally is important and, as such, this type of analysis is of interest to a small number of researchers. The analysis of the hemagglutinin variants is satisfactory but the neuraminidase analysis could be improved. There are some minor edits that would improve the manuscript.

Suggested minor edits:

Line 61-62           “…conducted a molecular analysis of viruses isolated from severe influenza…”

Lines 84, 85,87, 98           missing µ symbol for measurements (µL or µg)

Line 101               “…dilution to display…”

Line 113               “…characteristics…” (plural)

Table 1                 “Number of cases (n)” not %

Line 140               “… other hand, viruses with genetic changes in the NA gene may produce a neuraminidase protein that is resistant to the drug, leading to…”

Line 159               Were the influenza B cases associated with patients of all ages or with the younger patients? Influenza B infections usually occur in children or young adults and it is not clear if that is the case in Thailand.

Lines 190-197     Please check the labelling of amino acids at position 200 and 202 as the text is inconsistent and does not match the figures.

Line 194               “…mutation (S202P) was observed at the receptor…”

Line 204               “…tree from the concatenated segments from the 7 influenza A/H3N2… was constructed.”

Line 216               “…E78K/G…” (uppercase)

Lines 214 & 217 Please consider using the verb “have” or “contain” rather than “hold”.

Line 262               the sentence “According to the results of the phenotypic study” doesn’t make sense.

Line 311               “markers” rather than “makers”

Lines 306-313     The discussion of the susceptibility to oseltamivir should be presented as a separate paragraph. Comparison of the IC50 values of the viruses analyzed in this work with that of other publications would be useful. The references the authors include indicate that the IC50 values presented in Table 2 are typical of influenza B viruses (Burnham et al states that the baseline IC50 for influenza B viruses is typically 10-fold higher than that for influenza A viruses). Inclusion of influenza B viruses with high and low IC50 values (as was done for the H1N1 and H3N2 strains) would be a useful way to convey this to the reader.

The manuscript mentions the differences in the NA gene in line 239 but there is no discussion about how, or if, these may affect susceptibility to oseltamivir. Line 313 states that a clear understanding of the NA molecular markers is needed. If, as it appears, the IC50 values for the variants identified in this study is similar to other influenza B viruses then the authors should conclude that these variants have no significant role in neuraminidase inhibitor susceptibility and state that.

Reviewer 2 Report

The manuscript presented by Boonnak et al. entitled " Molecular characterization of seasonal influenza A and B from hospitalized patients in Thailand in 2018-2019" proposes a descriptive analysis of influenza A H1N1 and B strains isolated from severe cases. 7 strains of H1N1, 7 strains of H3N2 and 6 strains of influenza B have been sequenced. Phylogenetic trees were used to classify the strains in the existing clades. The three-dimensional structures of HA with the identified mutations were analysed. A brief assay of the susceptibility to oseltamivir of the strains studied was also carried out.

The interesting point of this manuscript is the detailed molecular description of the isolated strains and their placement in the phylogeny of the circulating strains between 2018 and 2019.

The weak point of this study is the oseltamivir sensitivity assay which must be done on at least 3 viral productions of each lineage with a statistical analysis to validate the robustness of the IC50.

Overall, although the results obtained enrich the database collected on the H1N1, H3N2 and B influenza strains, further characterisation would be required.

Others points:

- line 91 and HA assay: this is not a viral titer but a hemagglutination titer.

- HA titer results are not presented.

- Check the units in the materials and methods section

- line 168: are the viral isolates studied representative of the circulating strains in Thailand?

-Line 200: Are the mutations identified in the H1N1 NA (for exemple) documented elsewhere? What is their impact on oseltamivir resistance?

- Do the mutations identified in HA and NA have an impact on the fitness of the virus, on the HA/NA balance?

Reviewer 3 Report

Molecular characterization of seasonal influenza A and B from hospitalized patients in Thailand in 2018-2019

Kobporn Boonnak et al.

General:

The paper describes the molecular characterisation of 20 influenza viruses isolated from severe influenza cases. The paper reads well and the data are well presented.

Major comments:

A total of 30 patients were confirmed influenza cases, but in 10 cases the virus was not isolated in culture. Is this an expected isolation success and were there differences in the populations or time of isolation attempt (season)?

How many sputum samples of suspected influenza cases were collected?

Most of the patients had underlying conditions which increase the risk of severe influenza and it is not clear whether the case-severity is due to patient or virus characteristics. This should be addressed in the discussion under limitations of the study. Many of the isolates reported in the paper are closely related other isolates, were those isolated from severe influenza cases? If so this can be taken as an indication for higher virus pathogenicity.

Minor comments:

Line 159: How many patients were tested for influenza

Round 2

Reviewer 2 Report

The authors improved the manuscript and answered all questions.